# German Version of the Child Sexual Abuse Myth Scale (CSAMS-G): Translation, Expansion, and Construct Validation

**DOI:** 10.3390/bs15020143

**Published:** 2025-01-28

**Authors:** Lennart Bayer, Maike Cigelski, Justine Eilfgang, Elisabeth Barbara Kraus, Frieda Mensing, Simone Pülschen

**Affiliations:** 1Department for Interdisciplinary Collaboration in the Context of Sexual Violence, Europa-Universität Flensburg, Auf dem Campus 1, 24943 Flensburg, Germany; maike.cigelski@uni-flensburg.de (M.C.); justine.eilfgang@uni-flensburg.de (J.E.); frieda.mensing@uni-flensburg.de (F.M.); simone.puelschen@uni-flensburg.de (S.P.); 2Department of Psychology, LMU Munich, Akademiestr. 7, 80799 Munich, Germany; e.kraus@psy.lmu.de

**Keywords:** child sexual abuse myths, scale construction, confirmatory factor analysis, survivor attitudes

## Abstract

Research is needed on the myths regarding child sexual abuse in order to address commonly held misconceptions in persons training for professional careers in relevant fields for child protection. We present our translated, validated, and expanded Child Sexual Abuse Myth Scale (CSAMS-G). It was tested on a sample of 569 students studying either education, social work, law, or policing. Results of confirmatory factor analysis revealed a good model fit for our assumed factorial structure. Acceptable results on internal consistency were confirmed with McDonald’s ω. We also present the first results on the acceptance of child sexual abuse myths in our sample. Overall, myth acceptance was low, but a few exceptions were found, especially for the newly added items. We found group differences in factor scores for gender as well as between survivors and non-survivors of child sexual abuse.

## 1. Introduction

False beliefs and stigmatizations in the context of child sexual abuse (CSA) are commonly referred to as child sexual abuse myths (CSAMs). Belief in such myths is associated with adverse reactions when helping abused children and youths ([14]; [43]). [10] ([10]) developed the Child Sexual Abuse Myth Scale (CSAMS) to assess this CSAM acceptance in different populations. The scale has been translated into different languages and validated in several countries ([8]; [11]). However, it has not yet been translated into German or validated in Germany. Nonetheless, there is a strong need to assess the CSAM acceptance of German professionals, because a lack of knowledge on CSA along with myth acceptance can undermine efforts to engage in CSA prevention or deal with suspected cases of CSA ([8]; [19]). Our article presents the development of the German version of the CSAMS as well as results of a first scale validation among German university students. Students in teaching, social work, law, and policing were chosen because they are future professionals in the field of child protection and we expect them to hold similar beliefs to those of professionals currently working in the field who will be addressed in future with the validated questionnaire.

### 1.1. The Function of CSAMs and the Importance of Research on Them

CSAMs are associated with a number of adverse attitudes, such as sexism ([8]; [12]; [32]), victim blaming ([13]), or a negative perception of CSA survivors and their credibility in criminal cases ([20]; [29]). These attitudes can have a negative impact on CSA survivors and inhibit their willingness to disclose their experiences if, for example, a trusted person does not believe them ([42]). CSAMs are part of a system of beliefs that justify CSA and shift blame from the perpetrator to the CSA survivor and/or their caregivers ([8]; [9]). Research reveals some overlap in the functioning of CSAMs and rape myths ([12]; [20]). Rape myths are false or stereotyped beliefs in the context of sexual violence in general ([12]), such as the belief that sexual assault would (partly) be the survivor’s fault if they were to walk home alone at night. Like rape myths, CSAMs can also perpetuate the belief that false accusations are common in cases of sexual violence or abuse. Studies clearly negate this ([5]), with false accusations estimated to be prevalent in about 2–5% of CSA cases ([36]).

Additionally, acceptance of CSAMs appears to catalyze adverse actions in the context of CSA. Police officers, for example, more frequently use inadequate interviewing techniques in interviews with CSA victims when they have a higher acceptance of CSAMs ([14]). [15] ([15]) conducted research on CSAM acceptance in criminal trials, and they stressed how adherence to CSAMs can influence the outcome of a CSA case—a finding found previously for rape myths and their acceptance ([16]). Going beyond the legal context, CSAM acceptance is associated with doubting CSA disclosures ([13]). Believing in CSAMs can lead to inadequate reactions to suspected CSA by teachers ([34]). For example, [21] ([21]) found that the majority of a non-representative sample of German teachers felt inhibited about reacting to suspected CSA or its disclosure due to misconceptions about the frequency of false accusations in CSA cases.

A couple of important insights have been gained that have enhanced the understanding of CSAM acceptance in recent years. First, there seems to be a gender difference in acceptance of CSAMs: men frequently prove to be more accepting of CSAMs than women. [8] ([8]), who developed and applied the Portuguese version of the CSAMS, found significant differences between men and women in three myths (x9, x11, x14) that diffuse the blame for CSA. [19] ([19]) used a self-developed questionnaire to assess myth acceptance and found men to be significantly more likely to think that CSA is more common among families with a low socioeconomic status and to think that children would make up stories of CSA. [29] ([29]) found that men put less blame on perpetrators, more often doubt the seriousness of CSA consequences, and think of survivors as being less credible. A German study by [21] ([21]) showed that two-thirds of teachers and teacher students in their sample from primary and secondary education adhered to the misconception that children frequently lie about CSA cases. [47] ([47]) used the CSAMS with a sample of German special education students who showed low overall CSAM acceptance. The factorial structure of the original CSAMS could not be validated in their study. Similarly, low CSAM acceptance was found in a survey among German special education teachers ([4]). [44] ([44]) used the original CSAMS to evaluate seminars for a sample of German university students and found low CSAM acceptance in pre-, post-, and follow-up testing. These studies underline the need for research on CSAM acceptance among school professionals in Germany, as well as the validation of a German version of the CSAMS. [30] ([30]) gave an adapted Spanish version of the CSAMS to a sample of teachers. They found that teachers who had education on CSA and special education teachers were more knowledgeable about the topic and less susceptible to CSAMs. Overall, their sample showed low myth acceptance, with most teachers showing no tendencies of victim blaming and a correct assessment of the credibility of a CSA disclosure. Nonetheless, many teachers in their sample still believed that there is a common profile of CSA perpetrators ([30]), and there still seems to be a belief that CSA survivors are to blame if they were wearing revealing clothes at the time of the assault ([31]).

Assessing CSAM acceptance makes it possible to initiate specific preventive measures to improve knowledge of CSA and attitudes towards CSA and CSA survivors. Hence, measuring CSAM acceptance is a necessary first step towards effective societal CSA prevention ([37]). As [13] ([13]) put it: “for CSA prevention efforts to be successful, it is essential that they target not only what scholars believe are myths but also assess gaps in knowledge that need to be addressed among professionals and laypeople”. [40] ([40]) further strengthened this argument with their demand for the development of effective preventive measures for adults and professionals in the context of CSA. Hence, it is necessary to assess CSAM acceptance in diverse populations and especially in professionals who frequently deal either with children in general or CSA in particular.

### 1.2. Using the CSAMS

The original version of the CSAMS by [10] ([10]) uses 15 items distributed across three factors (blame diffusion, denial of abusiveness, and restrictive stereotypes) and was validated in a South African sample. Blame diffusion consists of beliefs that people other than the perpetrator would be at least partly to blame in cases of CSA (e.g., x10: “Adolescent girls who wear very revealing clothing are asking to be sexually abused”). The factor of denial of abusiveness consists of CSAMs that downplay the effects of CSA on children and youths (e.g., x2: “Sexual contact with an adult can contribute favorably to a child’s subsequent psycho-sexual development”). Restrictive stereotypes are held by individuals who deny the factual reality of most CSA cases (e.g., x3: “Most children are sexually abused by strangers or by men who are not well known to the child”). To check the cross-cultural validity of the construct, the questionnaire was also tested in a South Korean and Swedish sample and retested in another South African sample by [11] ([11]). Additionally, the CSAMS has been translated into Spanish by [30] ([30]) and Portuguese by [8] ([8]). [30] ([30]), however, used an adapted version of the CSAMS by combining it with the Sexual Abuse of Males Perceptions Scale by [35] ([35]). They found significant group differences between teachers from regular schools and those working in special needs education. These recent studies found encouragingly low myth acceptance in their samples when using the original CSAMS ([8]; [29]). However, both studies found significant gender differences in CSAM acceptance, with men being more likely to adhere to CSAM than women. Another study assessing CSAM acceptance using a self-constructed measurement found that some myths (such as victim-blaming myths) are less accepted nowadays compared to myths focusing on the socioeconomic status of the families of CSA survivors or their relationship to the abuser ([19]). Overall, knowledge on CSA has improved recently and CSAM acceptance seems to be lower, especially in those CSAMs that were the focus of prior research ([40]). These findings led to us add four items to the original version of the CSAMS that depict more recent CSAMs (see Table 1).

Item x17 (Child sexual abuse is committed by a certain type of perpetrator) is an expansion of CSAMs that expects a perpetrator to fit a certain stereotype (e.g., a lonely, middle-aged man) and can be linked to the “restrictive stereotypes” factor in [10] ([10]). The other items come from research indicating that there might be a kind of “false confidence about child sexual abuse” among professionals in schools, child protection services, or the police ([21]). This false confidence can be seen in such topics as potential indicators for CSA (x19: There are some types of behavior in a child that are clear indicators of sexual abuse) or expectations regarding the ability of children to disclose experiences of CSA (x16 Children who have experienced sexual abuse are unable to express their experiences of abuse in words). Some aspects of such false confidence were also evident in [21]’s ([21]) study. Given that a deeper understanding of CSAMs has led to the assumption of “false confidence about child sexual abuse” as an additional factor, we expect that the substantive structure of the construct has changed. Therefore, we constructed a four-factor psychometric model based on the three-factor model of the original CSAMS ([10]). Our model consists of the following factors.

Trivializing child sexual abuse (x1 Sexual contact between an adult and a child, which is wanted by the child and which is physically pleasurable for the child, cannot really be described as being “abusive,” x2, x5, x8, x9, x12, x14). Trivializing child sexual abuse is a factor consisting of seven items. They all address myths that downplay the effects of CSA.Shifting responsibilities (x4 Children who act in a seductive manner must be seen as being at least partly to blame if an adult responds to them in a sexual way, x6, x10, x15). Shifting responsibilities consists of four items measuring whether participants believe in myths that shift responsibility for the perpetration of CSA from the perpetrators to either the victims or bystanders.Assumptions about perpetrators (x03 Most children are sexually abused by strangers or by men who are not well known to the child, x07, x11, x13, x17). This factor consists of five items. These all test for myths assuming that there is a certain type of perpetrator who is more likely to commit CSA. One new item addressing this factor was added to the CSAMS-G: Item x17.False confidence about child sexual abuse (x16 Children who have experienced sexual abuse are unable to express their experiences of abuse in words, x18, x19). This last factor consists of three items that are all new additions to the scale. Each item assesses the acceptance of myths about dealing with and detecting or disclosing CSA.

The CSAMS-G is the German and expanded version of the CSAMS developed by [10] ([10]). It consists of 19 items and measures CSAM acceptance. Participants rate the items on a 4-point Likert scale ranging from 1 (disagree) to 4 (agree). All items are coded to ensure that higher response scores indicate higher myth acceptance. In contrast to [10]’ ([10]) CSAMS, we decided to leave out a neutral response option so that participants had to decide whether or not they agreed with the statement. Although the CSAMS-G is not designed for a specific target group, it was tested on a sample of students studying education, social work, law, and policing in order to assess the myth acceptance of (prospective) professionals in these fields.

### 1.3. Study Aims

First and foremost, this study aimed to validate the expanded and translated CSAMS-G and the assumed factorial structure. To supplement the psychometric insights that were the main focus of this study, we aimed to gain insights into CSAM acceptance among German students who were studying to become professionals in fields that are relevant for child protection and law enforcement. We hypothesized that men would be more likely to accept CSAMs than women, because this gender difference is prevalent in many studies on CSAM acceptance ([8]; [29]). When assessing rape myths, [18] ([18]) found rape survivors to be less accepting of rape myths compared to those affected by other crimes. Though prior research has not shown this effect, it is suspected that exposure to sexist and myth-accepting attitudes in society might influence the self-perception of rape survivors ([7]). Due to the aforementioned overlap in the functioning of rape myths and CSAMs, we hypothesized that CSA survivors would be less accepting of CSAMs than other participants in our study. In short, we explored the following hypotheses:Men will show higher CSAM acceptance than women.CSA survivors will show lower CSAM acceptance than non-survivors.

## 2. Materials and Methods

### 2.1. Sample

We recruited university students attending German universities who were studying either education, social work, law, or policing. A total of 1350 students participated in the survey, with 697 completing the whole questionnaire. Controlling for study subject reduced the final sample to 569 students. Data collection started on 1 January 2023 and was completed on 31 March 2023. Our sample had a mean age of 24.52 years (range: 18–50 years). Table 2 presents further sociodemographic data:

### 2.2. Measures

This section presents the development of the CSAMS-G.

#### 2.2.1. Translation

The original version of the CSAMS was translated by three professional translators and two of the authors engaged in research on the topic. Following [2]’s ([2]) guidelines, our team discussed the translations in order to decide which translation of each item would best reflect its intended content. This process resulted in a preliminary translated version of the CSAMS-G. The original scale was also expanded to include four items addressing more recently relevant CSAMs.

#### 2.2.2. Cognitive Interviews

After translating the scale, cognitive interviews were conducted with six research assistants who were also studying teaching or policing—not only to ensure that each item was comprehensible and understood in the intended way but also to detect potential problems when completing the scale. These cognitive interviews were evaluated in a qualitative analysis ([26]), with the results being discussed in the research team to mitigate subjective influences. However, this did not result in any further changes to the items, because each seemed to be comprehended in the expected way.

### 2.3. Standard Pretest

A standard pretest was conducted with 12 subjects who checked the online scale for formal mistakes and for its usability under conditions as close as possible to those expected in the field ([26]). This resulted in minor changes to address spelling errors or technical flaws (e.g., filters were optimized because they did not work in the intended way).

### 2.4. Data Collection and Measurements

Data were collected with an online questionnaire using UniPark ([46])—a service compliant with German and European data protection laws (GDPR). Participants were recruited by contacting their universities via email and asking these to pass on the link to the questionnaire to students studying education, social work, law, and policing. All German universities teaching at least one of these subjects were contacted. Ethics approval for the study was granted by the Ethics Committee of the German Educational Research Association (GERA/DGfE; 09/2022/DGfE).

Participation in the survey was voluntary, and all participants gave informed consent based on information about the questionnaire provided prior to participation. Each participant was informed about the possible risks and emotional challenges of taking part in the survey. Additionally, participants were given contact data for our research team as well as for counseling centers specializing in CSA or sexual violence in general. They were informed that they could stop working on the survey at any time and that all data would be handled anonymously. By completing the survey, participants became eligible to voluntarily take part in a lottery with a chance of winning one of five EUR 25 book vouchers. The email addresses necessary for the lottery were collected in a separate survey to prevent any possibility of drawing conclusions about participants’ identities in the main dataset.

The CSAMS-G was handed to the participants as part of a larger set of instruments assessing six different topics together with sociodemographic questions. Sociodemographic data included age, gender, qualification, job, subject of study, study phase, number of semesters, personal experience of victimization, professional experience with cases of CSA, and use of support to deal with emotional and psychological challenges. Answering the question on personal victimization was voluntary. To ensure participants’ anonymity, their IP addresses were deleted from the dataset.

The CSAMS-G was part of a larger survey. The instruments included along with the CSAMS-G are listed in Appendix A. Except for the CSAMS-G, all instruments were presented in a randomized order. The CSAMS-G was always presented last because it was deemed necessary to add a subsequent disclaimer so that we would not be perpetuating any false beliefs by presenting them to participants. This disclaimer was shown after the CSAMS-G was completed and participants informed that the prior statements had to be regarded as myths for which there is no empirical support. The items of the CSAMS-G were also presented in a randomized order. Completing the whole set of instruments took the participants about 25 min.

### 2.5. Analysis

We performed all calculations with R version 4.3.2 ([38]). We tested for a multivariate normal distribution with the MVN package ([25]). Confirmatory factor analysis was performed using the lavaan package ([41]). We used the semTools ([23]) package to estimate the reliability of the subscales. We performed calculations for descriptive statistics with the psych package ([39]) and the performance package ([28]). We estimated factor scores with the lavaan package ([41]) and tested the group differences with the stats package.

### 2.6. Psychometric Modeling

We performed a confirmatory factor analysis (CFA) because this is the appropriate method for testing the model fit of an expected model ([33]; [45]). We evaluated the model fit using a cut-off value of higher than 0.95 for the comparative fit index (CFI), as proposed by [3] ([3]), as well as a combination of cut-off values of below 0.06 for RMSEA and below 0.09 for SRMR, as recommended by [22] ([22]). Additionally, we calculated a chi-squared goodness-of-fit test while also reporting the degrees of freedom associated with the model ([3]).

We assessed factor loadings of the CFA using weighted least squares means and variance-adjusted (WLSMV) estimation. We chose WLSMV because the data did not show a multivariate normal distribution and WLSMV is a robust nonparametric estimation designed for analyzing ordinally scaled data ([1]; [6]).

### 2.7. Internal Consistency

To check the internal consistency of the questionnaire, we calculated McDonald’s ω. We chose this measure because it has been found to assess the internal consistency of a measurement more accurately than Cronbach’s α ([17]).

### 2.8. Descriptive Statistics

We calculated descriptive statistics (mean, *SD*, median, min, max, range) for demographic data and single-item responses.

### 2.9. Group Differences

We assessed group differences for gender (non-binary participants could not be included in these calculations because they made up such a small proportion of the overall sample) and for CSA survivors versus non-survivors with Mann–Whitney U tests. We estimated the group differences using factor scores and then corrected for α mistakes using the Benjamini–Hochberg procedure.

## 3. Results

Because we did not find a multivariate normal distribution in our dataset, we used nonparametric tests for all calculations.

### 3.1. Psychometric Modeling

We tested the model fit of the CSAMS-G using a CFA with WLSMV estimation. All latent factors were allowed to correlate. Standardized values of the latent variables were used for the estimations. The model had a robust CFI of 0.955; a robust RMSEA of 0.029, CI 90% (0.023, 0.035); an SRMR of 0.074; and χ^2^ (146, 569) = 120.45; *p* = 0.941. Figure 1 presents the factorial structure of the model. Standardized loadings ranged between 0.17 and 0.50, being highest for Factor 3 (Assumptions about perpetrators).

#### Internal Consistency

Internal consistency was calculated for each subscale: Trivializing child sexual abuse (ω = 0.77), Shifting responsibilities (ω = 0.80), Assumptions about perpetrators (ω = 0.62), and False confidence about child sexual abuse (ω = 0.36).

### 3.2. Descriptive Statistics

Table 3 presents detailed information on descriptive statistics for the items. The mean scores for most items were low, indicating low overall CSAM acceptance. Higher means were found for items x7 and x13, as well as for the newly constructed items x16, x17, x18, and x19. Participants used the whole range of the Likert scale for each item.

### 3.3. Group Differences

#### 3.3.1. Gender

Gender differences were estimated using the Mann–Whitney U test. We found significant group differences in factor scores for three of the four factors. Men scored higher on “Trivializing child sexual abuse and shifting responsibilities,” whereas women scored higher on “False confidence about child sexual abuse” (see Table 4).

#### 3.3.2. CSA Survivors

Results of the test for group differences between CSA survivors and non-survivors are presented in Table 5. We found a significant group difference, with non-survivors being significantly more adherent to false assumptions about perpetrators.

## 4. Discussion

Before discussing the results on descriptive statistics and group differences, we shall focus on the CFA and our model.

### 4.1. Psychometric Modeling

This study aimed to validate the translated and expanded CSAMS-G. CFI, RMSEA, and SRMR were above the cut-off values proposed by [3] ([3]) and [22] ([22]), thereby indicating an acceptable-to-good model fit. The χ^2^ test was not significant (*p* = 0.941), indicating adequate global model fit. Internal consistency was tested by calculating McDonald’s ω. Because internal consistencies did not meet the commonly defined cut-off of 0.8, we do not suggest using the CSAM-G for individual case diagnostics. However, we do acknowledge the potential value of the factor scores for research purposes. Omega values for the subscales showed good internal consistency for the factors “Trivializing child sexual abuse” and “Shifting responsibilities”, as well as acceptable internal consistency for the subscale “Assumptions about perpetrators.” Only the subscale “False confidence about child sexual abuse” lacked internal consistency. We chose to retain the subscale because the items of this subscale were among the highest-scoring items of the measurement. The translated and expanded CSAMS-G works as intended, fitting the expected model and showing acceptable internal consistency, with the exception of the “False confidence” subscale.

### 4.2. Descriptive Statistics

When looking at the descriptive statistics of the CSAMS-G, we found very low overall CSAM acceptance in our sample. Nonetheless, some CSAMs still showed higher acceptance rates. Students seemed to be especially likely to accept the CSAMs that were added by expanding the scale: x16 and x19 were the only items in which the mean indicated more acceptance than non-acceptance of CSAMs. Additionally, there were a couple of the original items (x7 and x13) that scored higher. There seems to still be a need to apply information about CSA in the education of students who become professionals relevant for child protection, as demanded by [30] ([30]), though this need might not be as urgent anymore, considering the low results of our study.

### 4.3. Group Differences

Additionally, we tested two hypotheses. First, we expected men to show more myth acceptance than women. Our findings mostly converge with those of previous studies ([8]; [19]; [30]), because we found men to show significantly more myth acceptance for the factors “Trivializing child sexual abuse” and “Shifting responsibilities”. Surprisingly, for the newly added factor “False confidence about child sexual abuse”, we found women to be significantly more accepting than men, contradicting our hypothesis as well as previous research ([8]; [19]; [30]). Second, we expected to find less myth acceptance among CSA survivors compared to non-survivors. We found that CSA survivors do indeed show significantly less CSAM acceptance in the factor “Assumptions about perpetrators”. This corresponds to findings on rape myth acceptance reported by [18] ([18]). However, there were no other significant group differences in factor scores between CSA survivors and non-survivors, indicating a need for more research on these differences.

### 4.4. Limitations and Outlook

Although the CSAMS-G shows a good model fit and has proven its construct validity in a student population, further studies need to examine our assumed model structure in different populations. However, it does offer insights into the CSAM acceptance of our sample of future professionals whose duties will include child protection. Furthermore, the lackluster reliability of the factor “False confidence about child sexual abuse” needs to be addressed in a future adaptation of the CSAMS-G. We think that the new items provide valuable new information about what are the currently more common misconceptions regarding CSA. However, the items x18 and x19 might be misleading, because they are not formulated as strictly as they could have been (e.g., x18: “Child sexual abuse **always** involves the use of violence, and this leaves distinct physical traces”). The sample in which we tested the CSAMS-G is very homogeneous in the sense that we only recruited participants who are currently in higher education. This might have led to a restriction in range in the item responses and therefore not representing general CSAM acceptance in the German population of caregivers. There was also a bigger proportion of female than male participants in our sample. Future studies could aim to include more non-binary participants to check for gender differences in CSAM acceptance.

We designed the CSAMS-G to be used by research to evaluate CSAM acceptance in different settings and populations in Germany. Beyond that, it can be used to evaluate the knowledge of groups that will be attending CSA training courses or to test the efficacy of such courses in minimizing the CSAM acceptance of their attendees. By using the CSAMS-G, programs can be tailored to fit the educational needs regarding CSA and CSAMs in different populations. The CSAMS-G can also be used to examine biases in judicial decision-making in CSA cases, as done in a mock jury study covering rape myths by [27] ([27]). Combining the CSAMS-G with other measurements designed to survey rape myth acceptance, for example, the Modern Adolescent Dating Violence Attitude Scale ([24]) could be beneficial to examine relationships between these adverse attitudes.

## 5. Conclusions

The CSAMS-G offers a valuable addition to the investigation of CSAM acceptance in Germany. Despite the aforementioned limitations, we gathered valuable insights into CSAM acceptance in a diverse group of students who are all studying to become professionals in fields that are relevant for child protection. Even though CSAM acceptance is encouragingly low in our sample, some CSAMs are still more widely accepted. These CSAMs need to be addressed in educational programs and in-service training. Our instrument is suitable for research and the evaluation of training programs. However, future research should evaluate and test the CSAMS-G in different environments.

## Figures and Tables

**Figure 1 behavsci-15-00143-f001:**
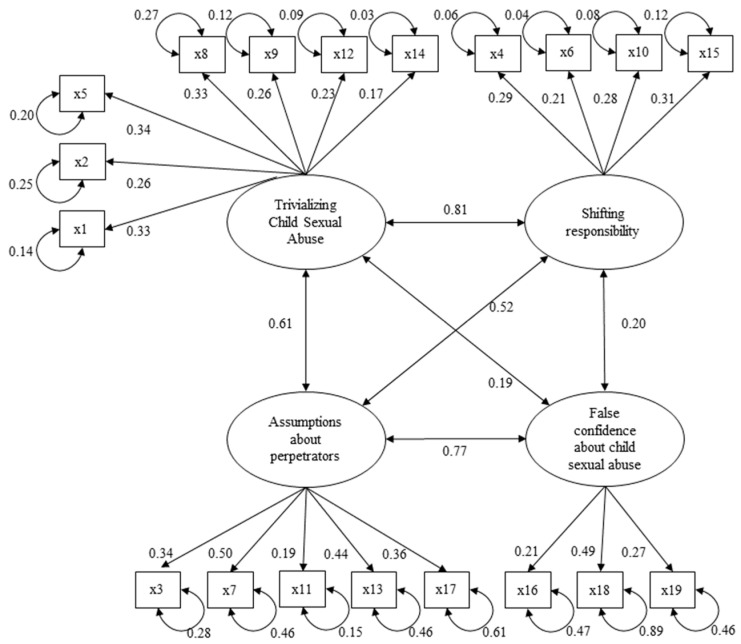
Factorial structure. Standardized loadings are indicated by arrows from factor to the items, error variances by half circles under the items, and factor correlations by double arrows between the factors for CSAMS-G.

**Table 1 behavsci-15-00143-t001:** New items.

Item
x16 Children who have experienced sexual abuse are unable to express their experiences of abuse in words.
x17 Child sexual abuse is committed by a certain type of perpetrator.
x18 Child sexual abuse involves the use of violence, and this leaves distinct physical traces.
x19 There are some types of behavior in a child that are clear indicators of sexual abuse.

**Table 2 behavsci-15-00143-t002:** Sample characteristics.

Gender	17.8% male 81% female 1.2% nonbinary
Subject	31.6% education 42% social work 10.5% law 15.8% policing
Personal victimization ^1^	10.4% victimized 88.4% not victimized 1.2% no answer

^1^ Personal victimization was evaluated by asking participants whether they were survivors of CSA. We made no differentiation between different types of sexual abuse and did not provide them with a definition for CSA. It was also possible to not answer this question.

**Table 3 behavsci-15-00143-t003:** Descriptive statistics for CSAMS-G.

Item		Mean	*SD*	Median	Min	Max	Range	Item Discrimination
x1	Sexual contact between an adult and a child, which is wanted by the child and which is physically pleasurable for the child, cannot really be described as being “abusive”.	1.209	0.496	1	1	4	3	0.49
x2	Sexual contact with an adult can contribute favorably to a child’s subsequent psycho-sexual development.	1.159	0.560	1	1	4	3	0.35
x3	Most children are sexually abused by strangers or by men who are not well known to the child.	1.469	0.630	1	1	4	3	0.46
x4	Children who act in a seductive manner must be seen as being at least partly to blame if an adult responds to them in a sexual way.	1.089	0.380	1	1	4	3	0.55
x5	Sexual contact between an adult and child that does not involve force or coercion and that does not involve actual or attempted sexual intercourse is unlikely to have serious psychological consequences for the child.	1.288	0.567	1	1	4	3	0.45
x6	A woman who does not satisfy her partner sexually must bear some of the responsibility if her partner feels frustrated and turns to her children for sexual satisfaction.	1.045	0.293	1	1	4	3	0.53
x7	Child sexual abuse takes place mainly in poor, disorganized, unstable families.	1.899	0.846	2	1	4	3	0.46
x8	It is not sexual contact with adults that is harmful for children. What is really damaging for the child is the social stigma that results once the “secret” gets out.	1.321	0.616	1	1	4	3	0.41
x9	Many children have an unconscious wish to be sexually involved with an opposite-sexed parent, which leads them to unconsciously behave in ways that make sexual abuse by that parent more likely.	1.163	0.439	1	1	4	3	0.46
x10	Adolescent girls who wear very revealing clothing are asking to be sexually abused.	1.100	0.400	1	1	4	3	0.50
x11	Children raised by gay or lesbian couples face a greater risk of being sexually abused than children raised by heterosexual couples.	1.115	0.424	1	1	4	3	0.49
x12	Boys are more likely than girls to enjoy sexual contact with an adult and are therefore less likely to be emotionally traumatized by the experience.	1.117	0.382	1	1	4	3	0.48
x13	Child sexual abuse is caused by social problems such as unemployment, poverty, and alcohol abuse.	2.022	0.805	2	1	4	3	0.42
x14	Children who do not report ongoing sexual abuse must want the sexual contact to continue.	1.036	0.245	1	1	4	3	0.58
x15	Older children, who have better understanding of sexual matters, have a responsibility to actively resist sexual advances made by adults.	1.181	0.464	1	1	4	3	0.48
x16	Children who have experienced sexual abuse are unable to express their experiences of abuse in words.	2.632	0.717	3	1	4	3	0.16
x17	Child sexual abuse is committed by a certain type of perpetrator.	1.961	0.856	2	1	4	3	0.33
x18	Child sexual abuse involves the use of violence, and this leaves distinct physical traces.	2.390	1.075	2	1	4	3	0.24
x19	There are some types of behavior in a child that are clear indicators of sexual abuse.	2.822	0.728	3	1	4	3	0.20

**Table 4 behavsci-15-00143-t004:** Group differences in factor scores between males and females. Significant results are indicated by an asterisk.

Factor	Males		Females		Test Statistics	*p*	*p* Adjusted
	Median	Range	Median	Range			
f1	−0.06	3.93	−0.19	3.15	27,931	0.003 *	0.004 *
f2	−0.1	3.26	−0.1	3.26	29,145	0.0001 *	0.0004 *
f3	0.06	2.88	−0.01	2.68	25,139	0.3	0.3
f4	−0.09	1.83	0.02	1.83	18,799	0.002 *	0.004 *

**Table 5 behavsci-15-00143-t005:** Group differences between CSA survivors and non-survivors. Significant results are indicated by an asterisk.

Factor	CSA Survivors		Non-Survivors		Test Statistics	*p*	*p* Adjusted
	Median	Range	Median	Range			
f1	−0.19	1.38	−0.11	3.93	13,418	0.2	0.267
f2	−0.1	0.68	−0.1	3.26	13,376	0.2	0.267
f3	−0.2	1.14	0	2.88	11,129	0.002 *	0.008 *
f4	−0.15	1.83	0.02	1.83	12,842	0.09	0.18

## Data Availability

The original data presented in the study are openly available on Zenodo at 10.5281/zenodo.14644857. Scripts for R alongside the questionnaire are available in the Appendix A.

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
