# Peer review of "German Version of the Child Sexual Abuse Myth Scale (CSAMS-G): Translation, Expansion, and Construct Validation"

_behavsci, 2025, doi:10.3390/bs15020143_

Round 1
Reviewer 1 Report
Comments and Suggestions for Authors
1. INTRODUCTION
This is clearly written, addresses most of the key references relevant to the research topic, and provides a compelling background and rationale for the study. However, a number of key references, relating to research on German participants, were not included. For example:
· Bienstein, P., Urbann, K., Scharmanski, S., & Verlinden, K. (2019). Prävention sexuellen Missbrauchs an Kindern und Jugendlichen mit Behinderung [Prevention of sexual abuse of children and adolescents with disabilities] . In: Wazlawik, M., Voß, HJ., Retkowski, A., Henningsen, A., Dekker, A. (eds), Sexuelle Gewalt in pädagogischen Kontexten [Sexual violence in pedagogical contexts]. Sexuelle Gewalt und Pädagogik, vol 3. Springer VS, Wiesbaden. https://doi.org/10.1007/978-3-658-18001-0_15
· Döring, N. & Walter, R. (25 Apr 2024): An Experiment on the Press Coverage of Child Sexual Abuse: Can Readers Differentiate Between Good and Bad Reporting? Mass Communication and Society, DOI: 10.1080/15205436.2024.2335344 https://doi.org/10.1080/15205436.2024.2335344
· Stück, E., Wazlawik, M., Stehr, J. et al. Teaching About Sexualized Violence in Educational and Clinical Institutions: Evaluation of an Interdisciplinary University Curriculum. Sex Res Soc Policy 17, 700–710 (2020). https://doi.org/10.1007/s13178-019-00427-8
· Verlinden, K., Scharmanski, S., Urbann, K., & Bienstein, P. (2016). Preventing Sexual Abuse of Children and Adolescents with Disabilities–Evaluation Results of a Prevention Training for University Students. International Journal of Technology and Inclusive Education, 5(2), 859-867. http://infonomics-society.org/wp-content/uploads/ijtie/published-papers/volume-5-2016/Preventing-Sexual-Abuse-of-Children-and-Adolescents-with-Disabilities-Evaluation-Results-of-a-Prevention-Training-for-University-Students.pdf
(pp.2 and 3 of 13; Line 96-99) At this point the authors maintain that:
“…it is necessary to assess CSAM acceptance in diverse populations and especially in professionals who frequently deal either with children in general or CSA in particular. To the best of our knowledge a study applying a standardized measure to assess myth acceptance in different professional groups is yet to be conducted in Germany”
While I agree with the general sentiments expressed by the author/s, I would like to direct the author/s attention to two studies (referenced above: Stück et al., 2020; Verlinden et al., 2016) that were conducted using German participants, with standardized measures being used to assess myth acceptance among different professional groups.
Concern 1 |
Action |
Response |
Key references relating to research on CSAMS are not included in the literature review |
Please examine the references provided and include those that need to be included |
|
Concern 2 |
Action |
Response |
Please revisit the author/s comment relating to research on CSAMS among different professional groups |
Please amend where indicated |
|
METHODS
Study methods are clearly described and appropriate. Regarding the translation process, it appears from the text that a number of independent translations were made and compared, What is not clear is whether a standard process of translation and back translation was followed to address semantic and content equivalence. A positive feature of the methodology was that cognitive interviews were conducted with a sub-sample of potential participants to assess the comprehensibility of scale items.
Concern 3 |
Action |
Response |
Were standard translation and backtranslation process employed in the translation process? |
If not, why not? If so, please describe more clearly |
|
RESULTS
The results section adequately addresses key research questions/hypotheses (sex differences and influence of past CSA experiences) with findings relating largely to what would be anticipated in terms of the extant literature. However, the four new items clustered largely within the
“False confidence about CSA” factor (X16, X18, X19) that had what many would regard as an unacceptably low McDonald’s ω coefficient ( ω = .36). Further, by the author/s self-acknowledgement some of the newly added items were not formulated as clearly as they could have been. Would appropriate rewording of these items x18 and x19 significantly change obtained estimates for the psychometric properties of the CSAMS-G? This issue needs to be more directly and comprehensively addressed by the author/s (maybe further research is indicated using reworded versions of additional questions?)
Issues relating to at least three of the four new items added to the CSAMS (x16, x18, x19) raise concerns in my mind regarding the incremental validity of the extended scale, over the original scale – with the issue of incremental validity not being addressed in the writeup.
Concern 4 |
Action |
Response |
Factor 4 (False confidence) demonstrated poor levels of internal consistency. |
Please provide a justification for retaining the “False confidence” factor given poor levels of internal consistency |
|
Concern 5 |
Action |
Response |
What is the incremental validity of using the extended scale |
Please assess the incremental validity of using the extended scale |
|
DISCUSSION
This section is well written, and provided me with no additional concerns
Author Response
Summary: Thank you for your extensive review and taking your time to read and comment our article! You can find our responses to your review in the following text. All changes made to the document are highlighted in red. Any parts which are deleted are currently crossed out so you can track exactly which changes have been made. For your information: We found an error with the corrected p-values of the group differences in gender. We corrected this error and also marked the passage in the table.
Comments 1: This is clearly written, addresses most of the key references relevant to the research topic, and provides a compelling background and rationale for the study. However, a number of key references, relating to research on German participants, were not included. For example [...].
(pp.2 and 3 of 13; Line 96-99) At this point the authors maintain that:
“…it is necessary to assess CSAM acceptance in diverse populations and especially in professionals who frequently deal either with children in general or CSA in particular. To the best of our knowledge a study applying a standardized measure to assess myth acceptance in different professional groups is yet to be conducted in Germany”
While I agree with the general sentiments expressed by the author/s, I would like to direct the author/s attention to two studies (referenced above: Stück et al., 2020; Verlinden et al., 2016) that were conducted using German participants, with standardized measures being used to assess myth acceptance among different professional groups.
Concern 1 |
Action |
Response |
Key references relating to research on CSAMS are not included in the literature review |
Please examine the references provided and include those that need to be included |
|
Concern 2 |
Action |
Response |
Please revisit the author/s comment relating to research on CSAMS among different professional groups |
Please amend where indicated |
|
Response 1: Thank you for pointing out the studies which were missing from our literature review. We have now included the studies by Stück et al. (2020), Bienstein et al. (2019) and Verlinden et al. (2016) to enhance the literature review of our study. The study by Döring & Walter (2024) shows an interesting facet of CSAM, however, we decided not to include the study because we found it to tackle topics beyond the scope of our article. Due to the additional literature, we decided to delete the section on missing research in Germany.
Comments 2: Study methods are clearly described and appropriate. Regarding the translation process, it appears from the text that a number of independent translations were made and compared, What is not clear is whether a standard process of translation and back translation was followed to address semantic and content equivalence. A positive feature of the methodology was that cognitive interviews were conducted with a sub-sample of potential participants to assess the comprehensibility of scale items.
Concern 3 |
Action |
Response |
Were standard translation and backtranslation process employed in the translation process? |
If not, why not? If so, please describe more clearly |
|
Response 2: Thank you for the feedback on our methods section. We did not employ backtranslation because we decided to test the quality of the translation via cognitive interviews. By analyzing the cognitive interviews in a similar sample, we were able to check whether each item is comprehensible and works as intended.
Comments 3: The results section adequately addresses key research questions/hypotheses (sex differences and influence of past CSA experiences) with findings relating largely to what would be anticipated in terms of the extant literature. However, the four new items clustered largely within the
“False confidence about CSA” factor (X16, X18, X19) that had what many would regard as an unacceptably low McDonald’s ω coefficient ( ω = .36). Further, by the author/s self-acknowledgement some of the newly added items were not formulated as clearly as they could have been. Would appropriate rewording of these items x18 and x19 significantly change obtained estimates for the psychometric properties of the CSAMS-G? This issue needs to be more directly and comprehensively addressed by the author/s (maybe further research is indicated using reworded versions of additional questions?)
Issues relating to at least three of the four new items added to the CSAMS (x16, x18, x19) raise concerns in my mind regarding the incremental validity of the extended scale, over the original scale – with the issue of incremental validity not being addressed in the writeup.
Concern 4 |
Action |
Response |
Factor 4 (False confidence) demonstrated poor levels of internal consistency. |
Please provide a justification for retaining the “False confidence” factor given poor levels of internal consistency |
|
Concern 5 |
Action |
Response |
What is the incremental validity of using the extended scale |
Please assess the incremental validity of using the extended scale |
Response 3: Again thank you for pointing this out. We agree that the "False confidence" factor shows low internal consistency and think that it should be subject to future research as mentioned in the limitations section in line 377. We decided to keep the factor because of the high average scores the items produced and added this explanation on line 348. At this point we also stressed that the "False confidence" factor is not showing acceptable internal consistency. In our opinion it is still valuable and adds incremental validity to the scale because it offers increased discriminability of the scores in populations that show low general myth acceptance.
The "False confidence" factor covers CSAM around talking with children or adolescents and their ability to disclose CSA experiences. It contains the items which had the highest acceptance rates of all items in this measure. This might show that knowledge of the traditional myths has increased in recent years due to campaigns and more media attention (e.g. #metoo). Removing it from the scale could potentially lead to ceiling effects for these populations and maybe even in the general population, since knowledge about CSA myths is expected to further increase in the following years.
Reviewer 2 Report
Comments and Suggestions for Authors
Congratulations on your work. Your research establishes an extremely relevant contribution to the development of new approaches to the myths of child sexual abuse, namely the possibility to improve of the original Child Sexual Abuse Myths Scale.
Still, I leave you one brief comment:
In lines 350 and 351 you mention that "expanding the scale – x16 and x19
were the only items in which the average indicates more acceptance than non-acceptance of the CSAM", however, if we look at the values ​​for question 7, they are close to the values of question 17. Question 13 even presents superior results to question 17. I suggest a better consideration on this data analysis, avoiding such a deterministic conclusion.
Thank you for the opportunity to read your article!
Author Response
Summary: First of all, thank you for taking your time and reviewing our article. Attached you can find our response to your comments. The changes stemming from this round of review are marked red in the new document. For your information: We found an error with the corrected p-values of the group differences in gender. We corrected this error and also marked the passage in the table.
Comments 1: Congratulations on your work. Your research establishes an extremely relevant contribution to the development of new approaches to the myths of child sexual abuse, namely the possibility to improve of the original Child Sexual Abuse Myths Scale.
Still, I leave you one brief comment:
In lines 350 and 351 you mention that "expanding the scale – x16 and x19 were the only items in which the average indicates more acceptance than non-acceptance of the CSAM", however, if we look at the values ​​for question 7, they are close to the values of question 17. Question 13 even presents superior results to question 17. I suggest a better consideration on this data analysis, avoiding such a deterministic conclusion.
Thank you for the opportunity to read your article!
Response 1: Thank you for your feedback! We added a short sentence on the items x7 and x13 in the lines 351 to 352. "Additionally, there were a couple of the original items (x7 and x13) which scored higher."
Round 2
Reviewer 1 Report
Comments and Suggestions for Authors
All requested revisions/corrections have been made to my satisfaction
Author Response
Thank you for taking your time and reviewing our article. Your revisions really improved our article!